# Isolation and Molecular Characterisation of Respirovirus 3 in Wild Boar

**DOI:** 10.3390/ani13111815

**Published:** 2023-05-30

**Authors:** Enrica Sozzi, Davide Lelli, Ilaria Barbieri, Chiara Chiapponi, Ana Moreno, Tiziana Trogu, Giovanni Tosi, Antonio Lavazza

**Affiliations:** Istituto Zooprofilattico Sperimentale della Lombardia e dell’Emilia Romagna “Bruno Ubertini” (IZSLER), Via Antonio Bianchi 7/9, 25124 Brescia, Italy; davide.lelli@izsler.it (D.L.); ilaria.barbieri@izsler.it (I.B.); chiara.chiapponi@izsler.it (C.C.); anamaria.morenomartin@izsler.it (A.M.); tiziana.trogu@izsler.it (T.T.); giovanni.tosi@izsler.it (G.T.); antonio.lavazza@izsler.it (A.L.)

**Keywords:** *Respirovirus*, parainfluenza virus, wild boar, phylogenetic analysis

## Abstract

**Simple Summary:**

This article discusses the isolation of respirovirus 3 from a sample of femoral bone marrow from a wild boar carcass imported from Australia. The complete genome sequence was determined, and based on our results, the wild boar isolate may result from cross-species infection of wild boars with BRV3. The application of NGS techniques has allowed us to investigate and characterise the genome of this strain, which was initially isolated in 2004.

**Abstract:**

Paramyxoviruses are important pathogens affecting various animals, including humans. In this study, we identified a paramyxovirus in 2004 (180608_2004), isolated from a sample of the femoral marrow bone of a wild boar carcass imported from Australia. Antigenic and morphological characteristics indicated that this virus was similar to members of the family *Paramyxoviridae*. The complete genome phylogenetic analysis grouped this virus into genotype A of bovine parainfluenza virus type 3 (BPIV-3), recently renamed bovine respirovirus type 3 (BRV3), which also includes two swine paramyxoviruses (SPMV)—Texas-81 and ISU-92—isolated from encephalitic pigs in the United States in 1982 and 1992, respectively. The wild boar 180608_2004 strain was more closely related to both the BRV3 shipping fever (SF) strain and the SPMV Texas-81 strain at the nucleotide and amino acid levels than the SPMV ISU-92 strain. The high sequence identity to BRV3 suggested that this virus can be transferred from cattle to wild boars. The potential for cross-species transmission in the *Respirovirus* genus makes it essential for intensified genomic surveillance.

## 1. Introduction

Paramyxoviruses are a diverse group of viruses that cause systemic, exanthematous, respiratory, and neurological diseases that affect humans and a range of animals, including livestock species such as cattle, pigs, and poultry.

Since 2019, this family has been known as the most robust expansion of all genera. The latest taxonomy release of the International Committee for the Taxonomy of Viruses (ICTV) recognises 4 subfamilies and 17 genera that contain >70 species and includes global human and animal viral pathogens. Moreover, the *Paramyxoviridae* Study Group renamed all species in the family to comply with the ICTV-mandated binomial format (*Mononegavirales*: *Paramyxoviridae*). According to this rule, the genus *Respirovirus* within the subfamily *Orthoparamyxovirinae* includes: respirovirus laryngotracheitidis and respirovirus pneumonia (HRV1 and HRV3, formerly human parainfluenza viruses 1 and 3 [HPIV-1 and HPIV-3]), respirovirus bovis (BRV3, formerly bovine parainfluenza virus 3 [BPIV-3]), respirovirus caprae (CRV3, formerly caprine parainfluenza virus 3, CPIV-3), respirovirus muris (MRV, formerly Sendai virus [SenV]), respirovirus suis (PRV1, formerly porcine parainfluenza virus 1 [PPIV-1]), and respirovirus ratufae (GSqV) [1].

Paramyxoviruses are enveloped, single-stranded, negative-sense RNA viruses of 14.6–20.1 kb length. All respirovirus genomes encode six structural and two non-structural proteins in the following order: nucleocapsid protein (N), phosphoprotein (P), matrix (M), hemagglutinin/neuraminidase (HN), fusion (F), and large polymerase (L) protein. Two non-structural proteins, C and V, are expressed from the P open reading frame and are involved in innate immune response interference and particle infectivity [2]. Both F and HN proteins are involved in receptor binding, possess neutralising epitopes, and are the most genetically diverse viral proteins. Several novel paramyxoviruses have emerged in animals and humans in the last few decades, causing severe illness and death. Several of these paramyxoviruses are zoonotic, some of which—the Nipah and Hendra viruses—have high fatality rates in humans [3,4]. Swine is a primary reservoir of porcine respiratory virus type 1 (which was recently detected in the USA [5,6]), La Piedad Michoacan paramyxovirus (LPMV) [7], porcine rubulavirus (PoRV) [8], Menangle virus (MenV) [9], parainfluenza virus 3 (PIV3) [10], and porcine PIV5 (pPIV5) [11]. Cross-species transmission of paramyxoviruses from their hosts to swine has been reported, including the Nipah, Menangle, and Newcastle disease viruses and BRV3 [10,12,13,14,15]. In addition, Qiao et al. (2009) [16] isolated two swine paramyxoviruses, Texas-81 and ISU-92, from the brains of pigs that exhibited respiratory and central nervous system diseases in the 1980s and the 1990s from the South and North Central United States, respectively. Antigenic and molecular analyses have indicated that these two viruses are very closely related to bovine respirovirus type 3 (BRV3) in the genus *Respirovirus* [16].

In bovines, BRV3 infection results in asymptomatic to severe respiratory disease; however, no neurological disease has been reported. Based on genetic and phylogenetic analyses, three genotypes—A (BRV3a), B (BRV3b), and C (BRV3c)—have been described [17,18,19,20,21].

In recent years, with the advent of next-generation sequencing (NGS) technologies, the rate of discovery of viruses in domestic animals and wildlife has increased rapidly, and the generation of fully characterised animal viral genomes across diverse host species has improved our understanding of viral evolution and cross-species transmission. In 2004, a paramyxovirus was isolated from a sample (180608_2004) of the femoral marrow bone of a wild boar carcass imported from Australia in an Italian meat processing company in Forlì Province (Italy). In this study, we described the biological characteristics and performed a complete genome sequence analysis of this wild boar isolate, thus giving evidence of viral spill over between species based on the phylogenetic analysis of closely related virus sequences.

## 2. Materials and Methods

### 2.1. Sample

In 2004, the femur from a wild boar carcass imported from Australia was conferred to the IZSLER Virology Lab in Brescia (sample no. 180608_2004) for classical swine fever virus (CSFV—*Pestivirus C*) investigation, following national rules on the preventive control of imported meat. Bone marrow was collected from the femur and homogenised (10% *w*/*v*) in minimum essential medium (MEM; Gibco, Life Technologies, Paisley, UK) supplemented with antibiotics (1000 U/mL penicillin, 1 mg/mL streptomycin; Gibco, Life Technologies, Paisley, UK) and anti-mycotics (2.5 μg/mL amphotericin B; Gibco, Life Technologies, Paisley, UK). After centrifugation, the supernatant was stored at −80 °C.

### 2.2. Viral Isolation

Screening for *Pestivirus C* was conducted according to the World Organization for Animal Health (WOAH) standardised protocol [22]. Isolation was performed in rapidly dividing porcine kidney (PK-15) cells that were seeded, in the first passage, in culture tubes and 25 cm^2^ flasks; in the second passage, they were seeded in culture tubes, 25 cm^2^ flasks, chamber slides, and coverslips, with a 2% suspension of the supernatant in growth medium. As the growth of the virus does not cause a cytopathic effect (CPE), its eventual presence was demonstrated using immunostaining, which was carried out after two virus passages. This was conducted by examining the cultures for fluorescent foci using the fluorescent antibody test (FAT) using a homemade anti-*Pestivirus C* hyperimmune serum (data not published).

The supernatant of PK-15 cells inoculated with bone marrow homogenate was then inoculated into 24-well plates in sub-confluent monolayers of Madin–Darby bovine kidney (MDBK) cells, Vero cells, and primary swine kidney cells (SKI). Cells were maintained in minimal essential medium (MEM) with 1% L-glutamine 200 mM, 1000 U/mL penicillin, 1 mg/mL streptomycin, 2.5 μg/mL fungizone, and 10% foetal bovine serum (FBS), free of antibodies against bovine herpesvirus-1, bovine respiratory syncytial virus, and BRV3, and free of both virus and antibodies against bovine viral diarrhoea virus. The inoculated plates were incubated at 37 °C in 5% CO_2_, and after a 1 h adsorption period, the cell cultures were rinsed, and a maintenance medium was added. Cell cultures were observed daily for a CPE for 6 d. Two blind passages were made if no CPE was observed; the cell cultures were scraped and vigorously mixed with the culture medium and used for the inoculation of fresh monolayers.

### 2.3. Negative Staining Electron Microscopy (nsEM)

The supernatant from the cell culture showing a CPE was subjected to nsEM using the Airfuge (Airfuge, Beckman Coulter, Inc. Life Sciences, Indianapolis, IN, USA) method [23]. Samples were loaded into 175 μL test tubes in which were placed specific adapters for 3 mm carbon-coated Formvar copper grids for direct pelleting. Ultracentrifugation was performed for 15 min at 82,000× *g*. Then, the grids were stained with 2% phosphotungstic acid (NaPT) pH 6.8 for 1.5 min. The examination was performed using a Tecnai G2 Spirit Biotwin transmission electron microscope (TEM; FEI, Hillsboro, Oregon, OR, USA) operating at 85 kV. The observation was conducted at 13,500–43,000× for at least 15 min before being considered negative. The identification and recognition of the observed viral particles were based on their morphology.

### 2.4. Immunofluorescence (IF) Test

The virus isolate was inoculated into confluent PK15 cells in an 8-well chamber at multiplicities of infection (MOI) of 1.0, 0.1, and 0.01. For the direct FAT test, cells with a CPE were fixed with 99.5% acetone and incubated with FITC anti-BRV3 IgG. For indirect IF, cells were incubated with homemade anti-BRV3 monoclonal antibody (data not published) diluted adequately in phosphate-buffered saline (PBS) at 37 °C for 1 h. Then, cells were incubated with FITC anti-mouse IgG. In both cases, the slides were layered with buffered glycerin and observed under epifluorescence using an optical microscope.

### 2.5. Antigen Detection Enzyme-Linked Immunosorbent Assay (ELISA)

A homemade double antibody sandwich (DAS) ELISA, based on the use of monoclonal antibodies for BRV3 detection in biological samples and used in routine examinations, was performed on supernatant fluid from cell cultures showing a CPE (Appendix A).

### 2.6. Hemagglutination Assay (HA)

Culture supernatants from infected MDBK (50 μL) were serially two-fold diluted, titrated against 50 μL of guinea pig red blood cells (RBCs) by using V-bottom microtiter plates (Nunc), and incubated at 4 °C for 4 h. Then, we evaluated hemagglutination based on the appearance or absence of a red cell button and expressed the results as haemagglutinating units/50 μL (HAU/50 μL). We considered the last dilution at which haemagglutination was observed as the endpoint of haemagglutinating activity, and its reciprocal value was the expression of the number of haemagglutinating units or virus titres in the HAUs (Appendix A).

### 2.7. Reverse Transcription-Polymerase Chain Reaction (RT-PCR) and Sequencing

Infected MDBK cells were scraped off the plates and homogenised using three cycles of freezing and thawing. Viral RNA was extracted from 250 μL of supernatant using the QIAsymphony™ SP Instrument (Qiagen, Hilden, Germany), according to the manufacturer’s instructions. Oligonucleotide primers for BRV3 detection and identification described by Maidana et al. (2012) [24] were used to amplify a 328 bp segment of the consensus BRV3 Matrix (M) gene by RT-PCR using the commercial Qiagen One-Step RT-PCR kit (Qiagen). The PCR products were purified using the Qiaquick PCR Purification Kit (Qiagen). Sequencing reactions were performed with BigDye Terminator v3.0 kit (Applied Biosystems, Lennik, Belgium) and analysed with an ABI Prism 3730 DNA Analyser (Applied Biosystems).

#### Phylogenetic Analysis of Partial M Gene

M fragment nucleotide and predicted amino acid sequences of the wild boar isolate were edited and analysed with BioEdit version 11. We aligned this sequence together with other representative isolates of previously identified BRV3 and human respirovirus type 3 (HRV3), and to create alignments, we used Clustal W. Then, we used for phylogenetic analyses the nucleotide sequence alignments of the M gene fragment by using MEGA software, version 11. Finally, we investigated the phylogenetic relationships by bootstrap analysis (1000 replicates) using the maximum likelihood method with the Tamura 3-parameter model. Indeed, for nucleotide data analysis, we used rates among sites Gamma-distributed with invariant sites (G + I). Caprine respirovirus 3 (NC028362) was used as an outgroup.

### 2.8. Whole-Genome Sequence

By employing the MiSeq platform (Illumina, San Diego, CA, USA), we obtained the complete genome of the viral strain isolated on cell culture isolate (PK15). In particular, using an Illumina TruSeq RNA Library Preparation Kit v 2 and following the manufacturer’s instructions, we prepared the sequencing libraries. Reads of wild boar 180608_2004 were assembled de novo using CLC Genomic Workbench v.11 (QIAGEN, Milan, Italy) with an average coverage of 218. The full-length genome sequence of the wild boar isolate has been deposited in GenBank (accession number: OP341620). The complete genomes of three BRV3 strains (44306_2004, 87043_2004, and 300834_2004) isolated in our laboratory during routine diagnosis from bovine samples in 2004 on primary bovine embryonic kidney cells were also sequenced (accession numbers: OQ676820, OQ676821, OQ676822).

#### Phylogenetic Analysis

We aligned and compared the nucleotide sequence of the complete genome of strain 180608_2004 with the sequences available in GenBank (www.ncbi.nlm.nih.gov, (accessed on 16 June 2022)). This was performed using Lasergene software v 10.0 (DNAStar, Madison, WI, USA). In addition, to determine per cent identity, we used nucleotide BLAST (blastn) and protein BLAST (blastp) algorithms (http://blast.ncbi.nlm.nih.gov/Blast.cgi, accessed on 16 June 2022).

Phylogenetic analysis was performed using MEGA 11 software [25], and complete genome sequences of viral strains were compared with representative sequences of the *Respirovirus* genus obtained online (http://www.viprbrc.org, accessed on 25 August 2022) from the NIAID Virus Pathogen Database and Analysis Resource (ViPR) [26]. The complete genome BRV3 sequences were aligned with the corresponding reference sequences using MUSCLE from ViPR [26]. A maximum likelihood phylogenetic tree was constructed by applying a generalised time-reversible nucleotide substitution model and modelling among-site rate heterogeneity through a discretised gamma distribution with invariant sites (GTR + G + I) with 1000 replicates, identified using ModelFinder selection [27]. The differences between sequences were identified by computing the pairwise distance (P-distance) and computing the group mean distance in MEGA 11 [25].

## 3. Results

### 3.1. Morphological and Biological Properties of Wild Boar Isolate

Paramyxovirus was successfully isolated from the bone marrow of femurs obtained from a wild boar carcass imported from Australia. Isolate 180608_2004 produced a characteristic CPE (CPE) on PK15 cells, with scattered, rounded, refractory cells and small syncytia. A CPE was first visualised at the second passage, 72 h post-inoculation (DPI) (Figure 1A,B). The virus was then successfully isolated from MDBK, SKI and Vero cells and showed a CPE.

The supernatant of the infected cell culture preliminarily observed using nsEM revealed the presence of virus particles with morphological features indicative of the paramyxovirus family, revealing spherical to pleomorphic virions approximately 50–300 nm in diameter (Figure 1C). Intact virions were enveloped and densely packed with surface projections representing viral glycoprotein spikes. In addition, the nucleocapsids were visible and exhibited a typical “herringbone” pattern.

Both direct and indirect IF tests were positive in the cells infected with this isolate as well as the virological double antibody sandwich (DAS) ELISA assay.

Isolates from MDBK, PK15, SKI, and Vero cells agglutinated guinea pig red blood cells (RBCs). HA titres were 256, 128, 16, and 32 HAU/50 μL, respectively.

### 3.2. Molecular Characterisation RT-PCR and Sequencing

The M protein is the most conserved structural virion protein. A fragment of the M gene, consistent with the expected size of 328 bp, was amplified from the wild boar isolate 180608_2004 using RT-PCR. The nucleotide sequence of the virus identified in this study was closely related to that of bovine respirovirus 3 (Figure 2). Phylogenetic analysis showed that the identified 180608_2004 isolate was placed in the same clade as BRV3 (Kansas/15626/84 and Shipping Fever) and with two swine parainfluenza virus 3 strains—Texas-81 and ISU-92—isolated from encephalitic pigs in the United States in 1981 and 1992 [16]. Recently, it was reported that there are three BRV3 genotypes: A, B, and C [17,18,19,20,21]. However, two sub-genotypes of BRV3 were discernible in genotype A: the first genetic group represented by Kansas and Shipping Fever viruses and swine parainfluenza virus 3 81-19252 Texas-81; and the second genetic group represented by BRV3 910N, BP6121, and JCU strains, and swine parainfluenza virus 3 92-7783 ISU92 [2]. The 180608_2004 isolate was included in the first genetic group within genotype A. An analysis of the alignment of the matrix gene sequence revealed that it was related to the BRV3 strain Shipping Fever (AF178655) from the USA and SPMV strain Texas-8,1 with 100% identity both in nucleotide and deduced amino acid sequences. In contrast, it had 92.5% and 96.5% identity with the SPMV ISU-92 strain in nucleotide and amino acid sequences, respectively. Compared with HRV3, the wild boar virus had 80.4–81% identity with the M gene nucleotide sequence and 91–92% identity with the deduced amino acid sequence, showing higher levels of identity with BRV3 than with HRV3.

The nuclear localisation signal sequence—^245^KMGRMYSVEYCKQKIEK^261^—of the M proteins is highly conserved in BRV3 strains [28], and it was found to be also in the viral M protein of wild boar strain180608_2004, as well as in swine parainfluenza viruses.

The complete genome of 180608_2004 is 15456 nucleotides (nt) in length and encodes 6 genes, yielding the structural nucleoprotein (N), phosphoprotein (P), matrix (M), fusion (F), hemagglutinin-neuraminidase (HN) and polymerase (L) protein, as well as the accessory V and C proteins transcribed from the P gene.

The complete viral genome from this study, as well as full-length genomic sequences of representative *Respirovirus* genus members obtained from GenBank, including publicly available sequences of BRV3, PRV1, PRV2, HRV1, HRV3, CRV3, MRV, and GSqV, were aligned, and a phylogenetic tree was derived. The resulting dendrogram showed that the virus clustered in the BRV3a subgroup (Figure 3). Similarly, as shown in Figure 2, BRV3a appeared to form two subgroups, where the wild boar isolate was placed in the first subgroup with swine parainfluenza virus Texas-81.

BLAST analysis of the whole genome of the wild boar paramyxovirus, named wild boar parainfluenza virus 3 180608_2004, revealed that BRV3 had the most similar sequence, particularly in the Shipping Fever strain (accession number AF178655, 99.96%). In addition, the whole genome of WB/180608/2004 showed 98.2% nucleotide identity with the BRV3 Kansas/15626/84 strain, 98.2% with the swine parainfluenza virus 3 isolate Texas-81, 92.6% with the swine parainfluenza virus 3 ISU-92 strain, and 78.4% with the human parainfluenza virus 3.

Two surface glycoproteins—the hemagglutinin-neuraminidase (HN) and fusion (F) proteins—work together to mediate fusion into the target host cell during infection. HN is a tetramer that carries out several functions and serves as the receptor-binding protein for the virus by binding to sialic acids [29]. When HN is attached to a receptor, it activates F to a fusion-ready state. Once activated, F inserts itself into the target membrane and undergoes a series of conformational changes that lead to fusion. HN also stabilises the prefusion form of F before receptor engagement [30] and cleaves the receptor via its neuraminidase activity during viral budding. Such critical functions place the HN protein and HN–F fusion complex under significant selective pressure. The nucleotide sequences of the F and HN genes of 180604_2004 were determined, and the translation to protein sequences was obtained from Expasy translate online (https://www.expasy.org/search/translate, (accessed on 2 September 2022)). Individual genes are transcribed by a start–stop mechanism controlled by conserved sequences at the gene borders in members of the genus *Respirovirus*. The fusion (F) gene of the 180608_2004 isolate was 1869 nucleotides (nt) in length, with a single ORF of 1623 nt beginning at position 211, capable of encoding a 540 amino acid protein. It had 100% identity with both SPMV Texas-81 and BRV3-shipping fever (SF) strains, both in nucleotide and deduced amino acid sequences (Table 1). The F protein mediates the fusion of the virus and the cell membrane in paramyxoviruses. As Qiao et al. (2009) [16] reported, the F proteins of swine paramyxoviruses are essentially similar in structure and function to those of other paramyxoviruses, especially BRV3.

The hemagglutinin-neuraminidase (HN) gene was 1888 nt in length with a single ORF of 1719 nt beginning at position 74, which encodes a 572 amino acid protein. The wild boar 180608_2004 isolate shared 99.8% identity with the SPMV Texas-81 strain both in nucleotide and predicted amino acid sequences, and 99.8% and 99.7% identity with the BRV3-SF strain, in nucleotide and predicted amino acid sequences, respectively. Compared to HRV3, the wild boar virus had 75.2% and 70% identity in the HN gene nucleotide and deduced amino acid sequences, respectively (Table 1). Structure-based sequence alignment of 180608_2004 HN protein sequences with SPMV Texas-81 and BRV3-SF revealed only minor differences.

In addition, the P gene of 180604_2004 encodes two more viral proteins—C and V proteins—through the mechanisms of leaking scanning and mRNA editing, respectively, as described in Lamb et al. (2013) [2].

Sequence analysis showed that the 180608_2004 isolate had higher levels of identity with BRV3 than with HRV3. In addition, the 100% identity at the nucleotide level between the F protein of 180608_2004, the Texas-81 strain of PMVS, and the SF strain of BRV3 indicated that cross-species transmission might occur among different hosts.

## 4. Discussion

Currently, the family *Paramyxoviridae* includes eighty-six recognised species spread across four distinct subfamilies, the largest being the subfamily *Orthoparamyxovirinae* [31]. In addition to several important animal pathogens, this subfamily includes zoonotic pathogens such as the Hendra virus and Nipah virus, which can infect humans and other animals [32]. In recent years, the improvements in sequencing technologies, aided by next-generation sequencing (NGS), have resulted in the discovery of novel paramyxoviruses in pigs [33,34]. Among these, there are porcine *rubulavirus* and *Menangle* virus, classified in the subfamily *Rubulavirunae*, genera *Orthorubulavirus* and *Pararubulavirus*, respectively, and porcine respirovirus 1, in subfamily *Orthoparamyxovirinae*, genus *Respirovirus* [5,6,15,35,36,37]. Other paramyxoviruses associated with central nervous and respiratory disease in pigs also have been reported [10,38,39,40]. One is the novel porcine morbillivirus PoMV, within the Morbillivirus genus, considered the putative cause of foetal death and encephalitis in pigs [41].

In addition, metagenomic surveys allowed scientists to monitor viruses circulating in animal species and identify potential zoonotic threats [42,43,44]. While comparing novel virus sequences with known pathogens may help inform the risks associated with future spill over events, this type of in silico modelling based on viral sequences should also be complemented by the functional characterisation of such viruses based on virus isolation and in vivo studies elucidating pathogenesis.

In 2004, a sample of femoral bone marrow from a wild boar carcass, imported from Australia to an Italian meat processing company, was submitted to our laboratory for *Pestivirus C* screening as indicated by national rules on the control of meat importation. As a result, a parainfluenza virus 3 strain, 180608_2004, was isolated using PK15 cell cultures from the bone marrow. It is interesting to note that the first isolation was achieved quite promptly after two passages in the PK15 cell culture and that the isolate was then as easily isolated in three other cell lines (MDBK, SKI and Vero). This may indicate the considerably large capacity of this viral strain to attach to a variety of receptors and to infect different types of cells. However, it cannot be excluded that after the first isolation in PK15, it already became a more lab-adapted strain.

Morphological properties determined through EM observation and antigenic analysis using an indirect fluorescent antibody, sandwich ELISA, and haemagglutination assay were similar to those of members of the family *Paramyxoviridae*.

The progressive implementation of NGS techniques for identifying and characterising unknown and/or untyped pathogens has allowed us to investigate further and characterise the genome of this strain, which was initially isolated in 2004.

In the present study, the complete genome sequence was determined and deposited in GenBank under accession number OP341620. The viral genome of 180608_2004 is 15456 nt, which is typical for other paramyxoviruses. Six genes were found within the viral genome: nucleocapsid (NP), phosphoprotein (P), matrix (M), fusion (F), haemagglutinin-neuraminidase (HN), and large protein (L). Phylogenetic analysis of the full-length genome of wild boar paramyxovirus with other members of the *Paramyxoviridae* placed them in the same clade, along with BRV3. Phylogenetic analysis of the M gene sequence confirmed this grouping. This result is consistent with the classification of wild boar PIV3 isolates in the genus *Respirovirus* and *Paramyxovirinae* subfamilies.

Qiao et al. (2009) [16] identified two PIV3 strains isolated from pigs in the United States between 1980 and the early 1990s. These viruses were later determined to be BRV3, suggesting that they may be transferred from cattle to pigs but show both low pathogenicity and seroprevalence in domestic swine [37]. The host-specific amino acid differences between the Texas/1981 strain and BRV3 may result from an attempt to adapt BRV3 to a new species but without a real and consistent establishment in the population itself. In fact, according to the International Committee on Taxonomy of Viruses criteria for species demarcation, the diversity in glycoproteins and nucleoproteins is >7%, as has been shown for the ISU/1992 strain that can be considered a subspecies [38]. Based on our results, the 180608_2004 wild boar isolate may result from the cross-species infection of wild boars with BRV3. Host-specific sequence elements may contribute to the growth of the virus in heterogeneous hosts. Paramyxoviruses have the general characteristics of high replication efficiency, the ability to express foreign genes, and the capability to induce robust host immune responses [45]. Serum virus neutralisation assay or the hemagglutination inhibition (HI) assay may be used for serological application and serotyping if appropriate monoclonal antibodies are available. However, enzyme-linked immunosorbent assay (ELISA) is more suited for sero-surveillance and general routine diagnosis than neutralisation assays. These assays will be necessary for future studies to determine seroprevalence. The present study results are of interest for systematic studies of the epidemiology, genetic evolution, host adaptability, and pathogenic potential of these viruses in mammalian reservoirs and their possible spread or transmission into domestic animals.

## 5. Conclusions

*Paramyxoviridae* is a rapidly growing family of viruses that are known to affect a wide range of species, including humans, pigs, cattle, poultry, and companion animals. The application of NGS and the development of accurate analysis systems allow for the identification of novel paramyxoviruses, providing an extended evaluation of the diversity of these viruses. Indeed, in addition to various paramyxoviruses discovered in pigs [46], a new paramyxovirus, PRV-1, which was detected in dead pigs for the first time in Hong Kong in 2013 and successively reported in other countries, expanded the *Respirovirus* genus.

Wild boar parainfluenza virus 3 was only recently characterised by a complete genome sequence; therefore, it was possible to ascertain that it is likely a cross-species infection of BRV3 transmitted from cattle to wild boar, as was the case for SPMV isolated in the USA in the 1980s. BRV3 is mildly pathogenic to conventionally reared pigs, and a serosurvey conducted on swine farms showed negative results [37]. Cross-species infection with BRV3 has also been suggested in humans [47], lambs [48], and pigs [16,37].

Epidemic infectious disease outbreaks typically arise from viruses jumping between animal species, sometimes including humans. The *Paramyxoviridae* family contains a variety of highly infectious pathogens in humans and animals, and their potential to cross the species barrier and cause severe disease epidemics in new hosts requires continued surveillance. Our results, describing the biological characteristics and reporting the complete genome sequence analysis of a new wild boar isolate, provide additional insight into the diversity and evolution of paramyxoviruses, and give evidence of potential viral spill over events between wild and domestic species based on the phylogenetic analysis of closely related viral sequences.

## Figures and Tables

**Figure 1 animals-13-01815-f001:**
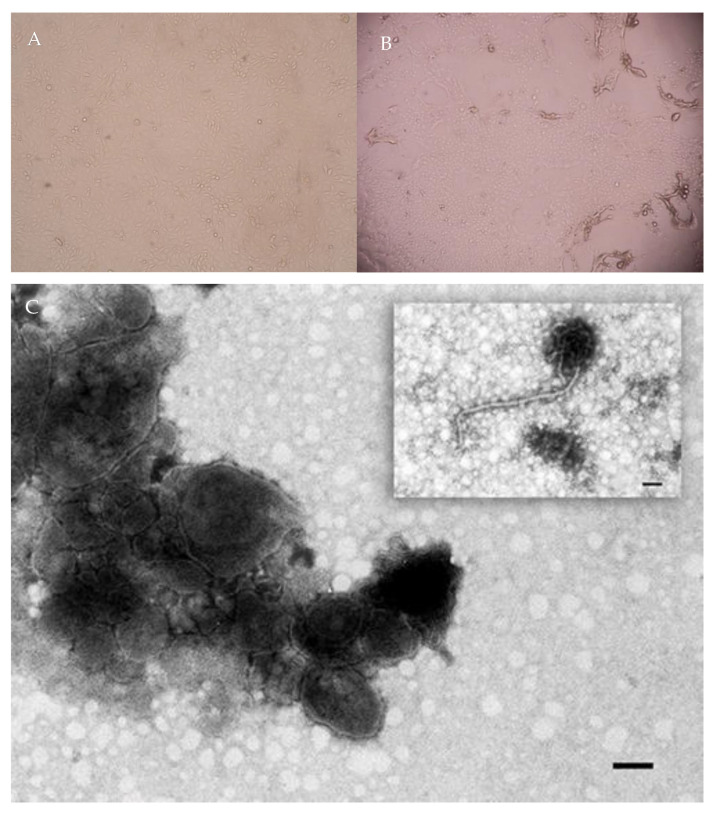
Microscopic examination of 180608_2004 isolated from wild boar. (**A**) PK15 cells (20× magnification). (**B**) PK15 infected cells showing lytic CPE at 72 h post-infection, as observed by Nikon Eclipse Ts2R inverted microscope (10× magnification). (**C**) Virus particles, morphologically indistinguishable from paramyxoviruses. Negative staining electron microscopy (nsEM), NaPT 2%. Bar = 100 nm.

**Figure 2 animals-13-01815-f002:**
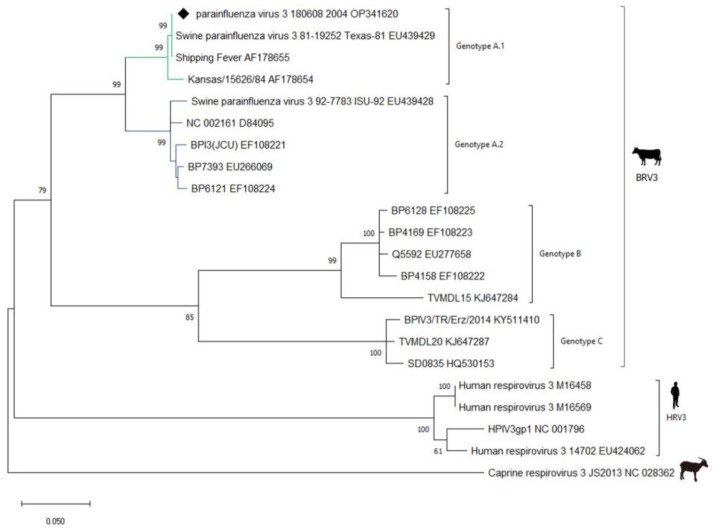
Phylogenetic tree generated based on the nucleotide sequence alignment of the virus’s BRV3 matrix (M) gene region obtained in the study and viral strains from GenBank. Molecular analyses were performed using MEGA 11 software with bootstrap analysis (1000 replicates) using the Maximum Likelihood method with the Tamura 3-parameter model and rates among sites Gamma distributed with invariant sites (G + I) for nucleotide data analysis. The wild boar 180608_2004 isolate is marked with black diamond. Published sequences are identified by strain and GenBank accession number. Caprine respirovirus 3 (NC028362) is an outgroup.

**Figure 3 animals-13-01815-f003:**
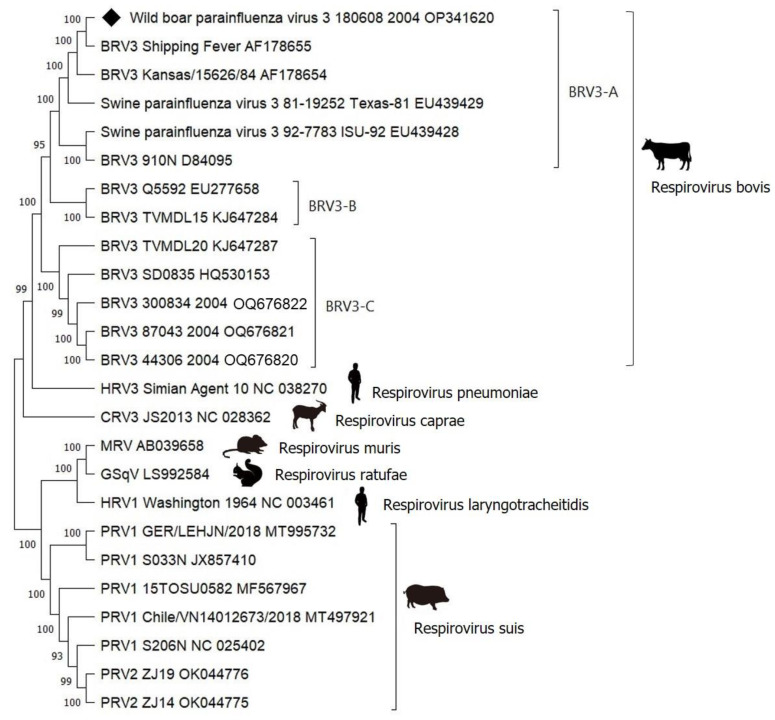
Phylogenetic tree based on the complete genome of wild boar parainfluenza virus 3 2004_180608 isolate and a selection of representative sequences of the *Respirovirus* genus. Molecular analyses were performed with MEGA 11 software using the maximum likelihood (ML) method based on the GTR + G + I model. Bootstrap values >70% are shown. Published sequences and references can be identified using the GenBank accession number. The black diamond indicates the novel sequence from wild boar obtained in the present study.

**Table 1 animals-13-01815-t001:** Nucleotide and amino acid identities between 180608_2004, SPMV, BRV3 and HRV3.

Virus	Strain	F		HN	
		Nucleotide (%)	Amino Acid (%)	Nucleotide (%)	Amino Acid (%)
180608_2004	SPMV-Texas-81	100	100	99.8	99.8
	SPMV-ISU-92	92	94.8	92	95.7
	BRV3a-SF	100	100	99.8	99.7
	BRV3a-NC002161	92.3	95	92.2	95.9
	BRV3c-SD0835	82.5	86.6	80.2	84
	BRV3b-Q5592	82.4	84.4	82.2	86.3
	HRV3-14702	78.4	79.5	75.2	70

## Data Availability

The data presented in this study are available on request from the corresponding authors.

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
