# Peer review of "Isolation and Molecular Characterisation of Respirovirus 3 in Wild Boar"

_animals, 2023, doi:10.3390/ani13111815_

Round 1

Reviewer 1 Report

In the manuscript "Isolation and molecular characterization of respirovirus 3 in wild boar”, Enrica Sozzi and al. described and sequenced a viral isolated from a femoral bone marrow from a wild boar. The phylogenetic analysis is very interesting but the molecular characterization could be improved, mainly by proving more data. Here are some comment that could improve the manuscript.

Major comments:

In general, the authors should include more citations.

Lines 205-208: The author should include the data for both DAS ELISA and the HA assay, at least in the supplementary section.

In the discussion, the authors should mention the novel morbillivirus in swine and discuss it (PMID: 34152961). Another paragraph discussing the relevance of the sequencing of the 180608_2004 after 4 to 6 passages (not clear) in 2 different cell lines should be added (e.g. PMID: 29970463). Would the author consider this virus as a lab-adapted virus already? Finally, would the author consider doing the reverse genetic of this virus to do further investigations?

Minor comments:

Line 35: Paramyxoviruses can be a lot longer that 16kb (e.g. Henipaviruses).

Line 44: The authors should also mentioned the non structural proteins.

Line 46: The classification has changed and the authors should add a more recent citation and adapt the text.

Line 140: What kit the author used for genomic RNA extraction?

Lines 193 and 201: The authors could make one figure out of the figures 1 and 2 (Figure 1.A and 1.B) since this is in the same cell type and at 72hpi even though this is not the same technic used.

Line 217: The authors should add a reference.

Line 243 and lines 292-294: The authors mentioned that the 6 genes can potentially encode for 9 proteins. This is true for the henipaviruses P gene that encodes for C, V and W non structural proteins but usually the other paramyxoviruses P gene encodes only for C and V non structural proteins. What about the isolate 180608_2004? Is the editing site in the P conserved compared to BRV3? The authors should include a reference (more recent than the number 1 from 2007).

Lines 266-269: The references 25 and 26 are very old and the function of proteins HN and F have been a lot better defined these past 10 years. The authors should update a bit this paragraph.

Reviewer 2 Report

Author comments

The research presents a genomic analysis of a paramyxovirus from wild boar. The paper is notable for potential evidence of viral spillover between species based on the phylogenetic analysis of closely related virus sequences and for the age of the samples, which were originally isolated in 2004 during screening for Pestivirus C, but only recently fully sequenced. I found the study to be well designed, and the paper well written. Some of the results could be expanded upon (see specific comments below) but overall, I have only a few minor comments which I hope will clarify the presentation of the work.

Line 221-222: “The 180608_2004 isolate was included in the first genetic group within genotype A.” Consider showing these subgroups on the figure, and/or switching branches so that group 1 is above group 2 to make this easier to interpret.

Line 222-226: “Analysis of the alignment of the matrix gene sequence revealed that it was related to the BRV3 strain Shipping Fever (AF178655) from the USA and with SPMV strain Texas-81 with 100% identity both in nucleotide and deduced amino acid sequences”. It looks like in Fig 3 that there are some differences between the study sequence and these references.

Figure 4: the host species icons add value to the figure, but species names should also be given especially for the wildlife species. Also, the same system would be nice in fig 3 as well.

Reviewer 3 Report

This study has a clear logic and reasonable design, which has good guiding significance for studying the spread and prevention and control of Wild board paramainfluenza virus 3 between hosts. However, there are still the following issues that we hope to improve:

1、189:Isolate 180608_ 2004 produced a characteristic CPE (CPE) on PK15 cells, similar to that of the BRV3 reference strip.

It is recommended to add images of the CPE of BRV3 and negative control in Figure 1 ,in addition please indicate the scale in the images.

2. Suggest improving the significance of this study to make it more comprehensive.
